# The Artisanal Fishing Sector in the Spanish Mediterranean: A Sector with a Long History and an Uncertain Future

Andrea Márquez Escamilla, Paloma Herrera-Racionero *, José Pastor Gimeno and Lluís Miret-Pastor 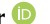

Research Institute for Integrated Management of Coastal Areas (IGIC), Universitat Politècnica de València, 46730 Gandia, Spain
* Correspondence: paherra@esp.upv.es

**Abstract:** The fishing sector is facing major economic and environmental challenges worldwide. However, at least in developed countries, it is also undergoing a major social crisis. This research has tried to quantify and validate this subjective feeling of deep crisis, based on objective and quantitative data referring to the Spanish Mediterranean. Indeed, the results show alarming data. Different scattered databases have been analyzed and it has been exposed that in only 15 years the number of vessels, power and tonnage has decreased by about 40%; as for fishermen, their number has decreased by about 30%, also revealing a serious problem of generational replacement, since in only 10 years, the average age of fishermen has increased by 3 years. A simple linear regression model shows that this downward trend in the number of boats and fishermen will continue at least in the short term. The data obtained invite a deep reflection on the future or even the very survival of fishing in the Spanish Mediterranean in the medium term. This reflection is not limited to this specific area but can be extrapolated to many other fishing areas.

**Keywords:** fishing; Spanish Mediterranean; generational replacement; young fishers

## 1. Introduction

Spain is the leading producer of fish in the European Union (18th in the world), with more than a thousand fish species of commercial interest. It represents an activity of enormous importance, perhaps not in economic terms (it barely represents 1% of GDP), but socially and culturally speaking. Its implications extend far beyond just the number of vessels or direct employment in the sector and provides food that is a fundamental part of the Spanish diet. Despite this, fishing seems to be increasingly overshadowed by other economic sectors such as construction, industry and especially tourism. Fishing is going through a critical moment that could put its very survival at risk.

The existence of pollution hotspots, illegal fishing, the introduction of non-native species and the effects of climate change raise major concerns regarding the survival of Mediterranean flora and fauna species [1,2], coupled with the increase in product prices and a significant increase in the demand for products, associated with population growth, have stimulated over-investment and, therefore, overexploitation of fishery resources, with 62.5% of fish stocks in the Mediterranean and Black Sea being overexploited [3].

All these pressures have had a significant effect on coastal fisheries and their profitability, leading to a lack of generational renewal that threatens the sustainability of the sector and rural fishing areas [3,4]. This reflects a trending inactivity in traditional Mediterranean fleets (36% of the total no longer active), with worsening future prospects.

Exacerbating this already difficult situation in the Mediterranean are other factors that have not been a priority in fisheries management: the lack of generational replacement, the limited possibility to compete with deep-sea fishing and aquaculture or the problems associated with gender roles [5–8].

Moreover, as is the case in the European landscape as a whole, fishing communities have suffered the consequences of the multiple reforms of the Common Fisheries Policy

(CFP), without being supported by fisheries policies, which, on the contrary, have focused their attention on environmental and economic problems, distancing the social dimension from management systems and bodies. In fact, fishing communities play a very minor role in the decision-making processes that ultimately influence their lives and futures [1].

From an environmental point of view, the different European fisheries policies and their management systems have tried, although perhaps not with the expected success [9,10], to deal with the problems of overexploitation and/or fishing overcapacity by focusing on the Maximum Sustainable Yield (MSY) criterion. In the Mediterranean, which has the highest rates of unsustainable exploitation in European waters [1,10,11], measures have been taken that include the Multi-Annual Management Plans (MAPs), recently adopted by Spain, France and Italy [12]. These include, among other provisions, the reduction in fishing by 40% in the period 2019–2024.

From an economic point of view, the European Commission is promoting blue growth as a strategy for economic growth centered on oceans and seas. Defined by the EU as a "long-term strategy to support sustainable growth in the marine and maritime sectors as a whole", it focuses on harnessing the untapped potential of Europe's oceans, seas and coasts for jobs and growth. Its main areas of action are aquaculture, coastal tourism, marine biotechnology, ocean energy and seabed mining. The fisheries sector has been omitted from the Blue Growth Strategy's priority areas [1] since, owing to the state of the fish stocks, economic growth potential is limited [13]. Furthermore, the parallels that can be found with the Green Revolution [2] in agriculture [14] should alert us to the mistakes made, notably the reduction in biodiversity and the species it condemned to extinction [15].

In any case, one of the stated objectives of fisheries policies is to increase employment and to diversify the economy within a transitional fisheries model towards sustainability [16]. Achieving such sustainability requires the integration and balance of the social aspect in management and governance processes. However, its absence, identified as one of the main problems causing the sector's labor recession, is more than palpable in fisheries measures, which are mainly ecological and economical in nature [17,18]. This not only leaves to one side the social considerations that inevitably play an essential role, but also the fishers themselves, their interests, opinions and knowledge [19], which are essential in fisheries management [7,20]. As a result, small-scale fisheries, the majority in the Spanish Mediterranean, are seeing their social resilience endangered, as they have less and less chance to recover in the face of the major political, social, economic and natural changes to which they are subjected.

Because of all the above circumstances, a current feeling of serious crisis coupled with even worse prospects for the future seems to have settled among fishing communities in most of the developed countries, or at least in Europe and Spain. However, little attention seems to have been given to the social situation compared to the environmental or profitability problems linked to fishing. This may be due to multiple causes such as the scarcity of reliable data, the multiplicity of situations, the scarce labor and economic importance of fishing in the modern economy, etc. However, little by little, a subjective(?) impression of pessimism about the future of fishing seems to be spreading, at least in certain areas. Quantifying the fishing crisis, at least in Spain, is not easy, since there are few social and labor data scattered among different databases and institutions. In this research, we will analyze and group data, trying to contribute objective and quantitative data to this discussion. We will focus on a wide area such as the Spanish Mediterranean, a territory with a long fishing tradition but where the current fishing crisis has plunged the sector into a deep depression where even the very future of the sector is beginning to be a matter of debate. The aim of this paper is to analyze the evolution of different socio-economic indicators in order to establish quantitatively the magnitude of the current socio-labor crisis in the sector and to establish what trend is emerging. Once the data have been analyzed, key discussions for the sector are raised, such as generational replacement or the future of fishing itself.

## 2. Materials and Methods

Our study focuses on the Mediterranean coast of the Iberian Peninsula, which extends more than 2000 km through Catalonia, Valencia, Murcia and Eastern Andalusia. Sitting opposite this coastline is the archipelago of the Balearic Islands with some 1428 km of coastline of great ecological value (Figure 1).

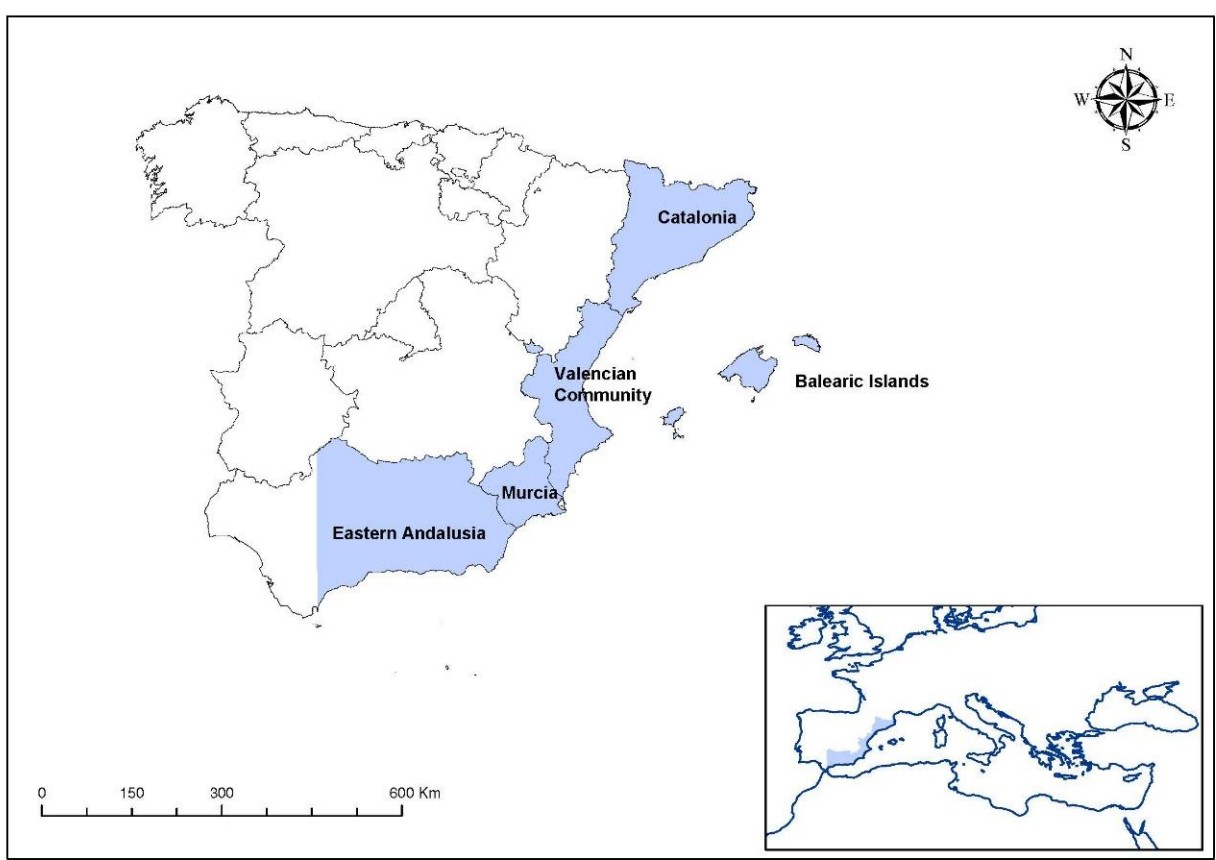

**Figure 1.** MAP 1 Study area.

A major difficulty in characterizing the current situation of fishing in the Spanish Mediterranean is the availability of data. The scarcity and imprecision of this data or the heterogeneity and dispersion of the databases relating to the sector are not exclusive to the Spanish State and explain the low representativeness of this activity in the reports that member states send to official bodies such as the FAO [21–23]. In our case, in the various sources consulted—the National Institute of Statistics INE), the Social Marine Institute (ISM) or the Ministry of Agriculture, Fisheries and Food (MAPA)- there is no information, at least published, with extensive time series and disaggregated by similar categories. Moreover, in most cases, it is impossible to find information relating exclusively to the fishing sector because it is often presented integrated with other activities of the primary sector, such as agriculture and livestock, and when it is separated, for example, in the ISM, we do not know the work associated with each fishing gear. Other times, the activity is specified, but not the geographic area where it is practiced.

In an attempt to minimize these shortcomings, through various processes of access to public information, we have opted to directly use the data compiled by the ISM on affiliations to the Marine Special Regime (REM) according to contribution groups, genders and ages for the period 2009–2020 and have combined them with the databases available, both at a national level (MAPA, INE) and from different regions.

As far as possible, we worked with annual data over a period of 14 years (2006–2020). Descriptive statistics were used to analyze the relationship between the selected variables

and their trends. In order to generalize and give robustness to the interpretations of the results, statistical techniques for hypothesis testing were applied using SPSS Statistics.

On the other hand, although the volume of data handled is not large enough to work with time series, the marked trend shown by the variables has allowed us to make a short-term prediction of the number of boats and fishermen by means of simple linear regressions.

Compared to similar results obtained with an exponential model, a simple linear regression has been the chosen model. The reasons are diverse. On the one hand, simple linear regression allows us, with little information, to show if there is an association between the variables, which is what we intend, without the need for it to be strictly causal, and therefore the predictions must be taken with some caution. On the other hand, in addition to being easy to interpret and explain, the results obtained in this study are good and statistically significant.

It is also important to note that we know the ARMA model would be another option to consider when making the forecast. However, it requires that the series be stationary, a condition that does not occur, which complicates its application in this case. This is the reason why it has been discarded.

## 3. Results

The objective is to quantitatively analyze the situation of the Spanish Mediterranean fishing sector based on different indicators. One of the most evident is the variation in the number of vessels. During the period between 2006 and 2020, the Mediterranean fishing fleet decreased by about 40%, from 3908 to 2341 vessels. These decreases have affected both minor arts (which constitute 62.7% of the total fleet), as well as purse seine, trawl and longline (Figure 2).

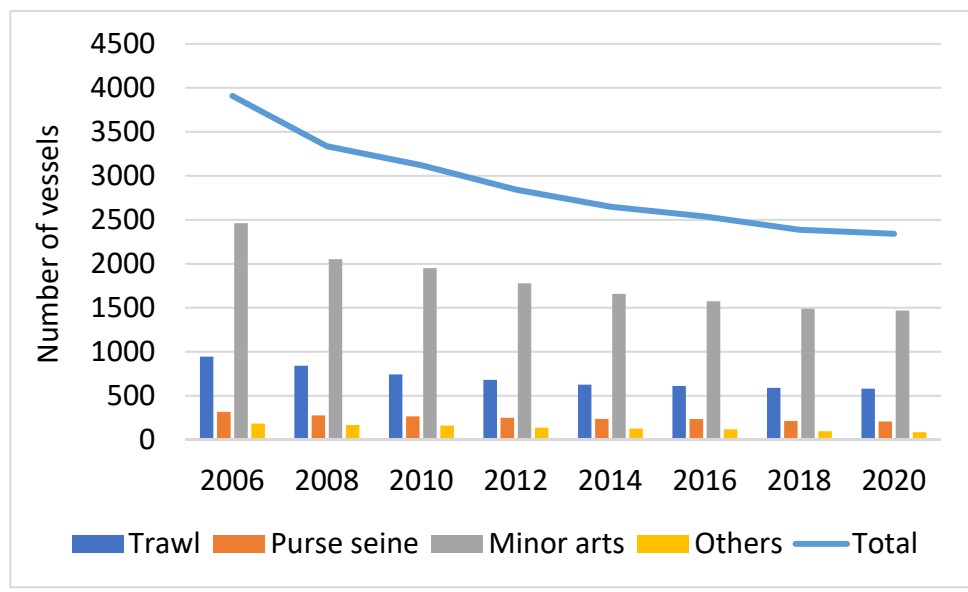

**Figure 2.** Evolution of the number of fishing vessels (2006–2020) [3].

This downward trend in the number of vessels is also reflected in the tonnage (GT) and power (hp) of the fleet (Figure 3), indicating a marked reduction in fishing capacity [25].

Since 2006, power has decreased by 38.5%, from 440.058 hp to 270.662 hp in 2020 (Figure 3). For its part, the tonnage has decreased from 80.341 to 50.446 GT, a decrease of 37.2% in fourteen years. If we divide these numbers by the number of boats in each sector, we obtain the Table 1:

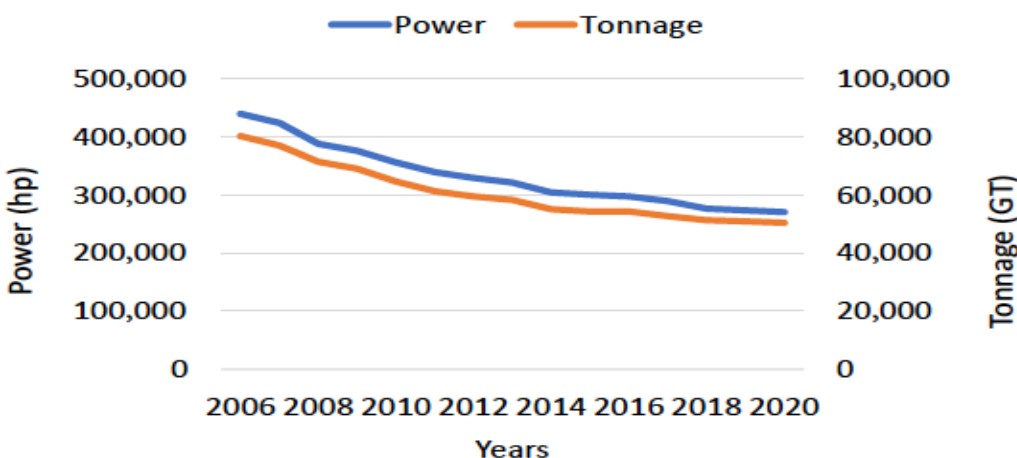

**Figure 3.** Evolution of the power and tonnage of the fishing fleet in the Spanish Mediterranean (2006–2020). Annual average values [4].

**Table 1.** Power (hp) and tonnage (GT) per boat by gear in the Spanish Mediterranean [5].

|  | Trawl | | Purse Seine | | Longline | | Minor Arts | |
|---|---|---|---|---|---|---|---|---|
|  | Tonnage GT Per Boat | Power hp Per Boat | Tonnage GT Per Boat | Power hp Per Boat | Tonnage GT Per Boat | Power hp Per Boat | Tonnage GT Per Boat | Power hp Per Boat |
| 2006 | 60.26 | 263.56 | 36.99 | 242.64 | 17.86 | 98.17 | 3.40 | 39.00 |
| 2010 | 59.55 | 254.01 | 40.06 | 252.95 | 16.88 | 102.03 | 3.66 | 43.11 |
| 2015 | 58.49 | 247.13 | 41.99 | 256.64 | 18.50 | 109.27 | 3.85 | 46.16 |
| 2020 | 57.96 | 240.82 | 43.38 | 252.55 | 21.32 | 108.35 | 4.08 | 47.17 |

Although there are fewer boats (and less tonnage and total power), the conclusions that can be drawn on the average size per boat are not so obvious. In fact, on the one hand, a decrease in the size and power of the trawl boats is observed (the average tonnage has decreased from 60.26 GT to 57.96 GT and the average power from 256.57 hp to 240.82 hp). This shows us that the decrease in the trawl fleet has mainly affected the larger and more powerful boats, reflecting the pressure of European policies on trawling and/or the evolution of costs. However, on the other hand the opposite seems to have occurred for the other gears. If we focus on the minor arts, it can be observed that in the same period, the boats have increased their average tonnage (from 3.4 to 4.08 GT) and their average power (from 39 to 47′17hp). In this case, it is the smaller boats that are disappearing, probably for not reaching the minimum turnover needed to breakeven.

The analysis of catches and their value shows a more irregular trend (Figure 4).

Although visually the behavior of both variables is similar, we observe that the value in the fish market fluctuates more markedly because it responds to other factors such as economic cycles or varying degrees of competition from other fishing areas [27].

The downward trends are also reflected in the number of fishers (Figure 5).

From 2009 to 2020, average annual affiliations decreased by 30.0%—at an average annual loss rate of −3.3%—which means that, for this last year, 3628 fishers disappeared in the Spanish Mediterranean.

The average age of fishers affiliated to REM in 2020 was 45 years old. The fishing situation, in terms of age distribution of workers, at a national level is similar, in fact, the average age is 45 years, the same as in our study area.

In order to obtain an idea of the level of aging of the fishing population, we compared the affiliations by age brackets to the REM with the total number of affiliations in all sectors (Figure 6).

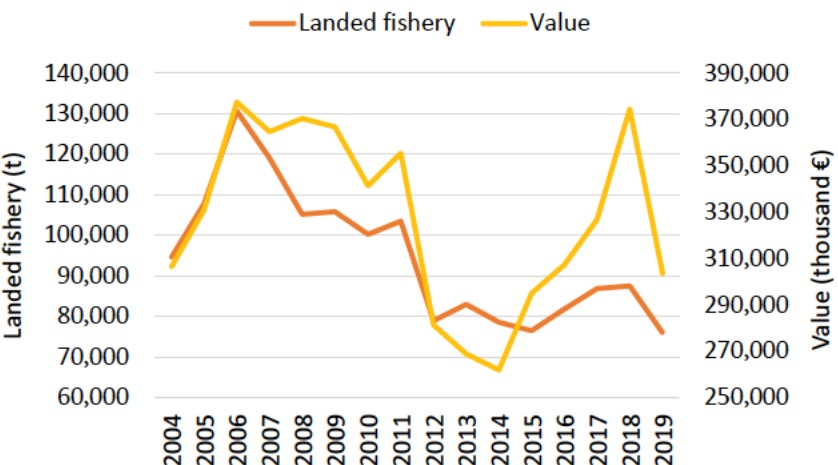

**Figure 4.** Evolution of the landed fishery (live weight) and its value in the Spanish Mediterranean (2004–2019). Annual average values [6].

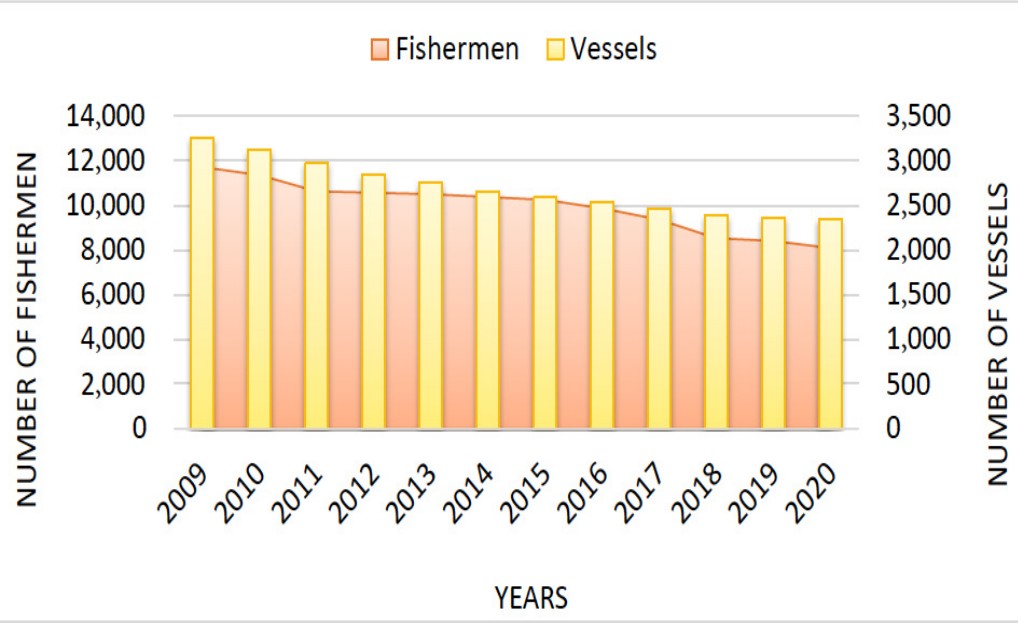

**Figure 5.** Comparison of the evolution of the total number of vessels and the average number of fishermen affiliated to REM in the Mediterranean [7].

The average age of fishermen is 45 years, whereas the average age of workers in the general labor market is 44 years. Although this difference does not seem very significant, it is, since it must be taken into account that Spanish law allows fishermen to retire 10 years earlier than other workers [8]. This is clearly observed when the population pyramids are compared.

The segment with the most fishers is between 50 and 59 years of age, while for the general population, it is between 40 and 49.

If this data is compared with that of 2009, we can see that over the last 11 years, there have been significant changes. Figure 7 shows a clear aging of the sector, increasing the proportion of fishers over 50 years of age (>12% of the total) and decreasing for the rest of the age groups, compared to 2009. In fact, the average age of fishers has increased throughout this period by 3 years: in 2009 it was around 42 years in the Spanish Mediterranean. With access to the historical series of these affiliations, deeper research could be made into the aging of the sector.

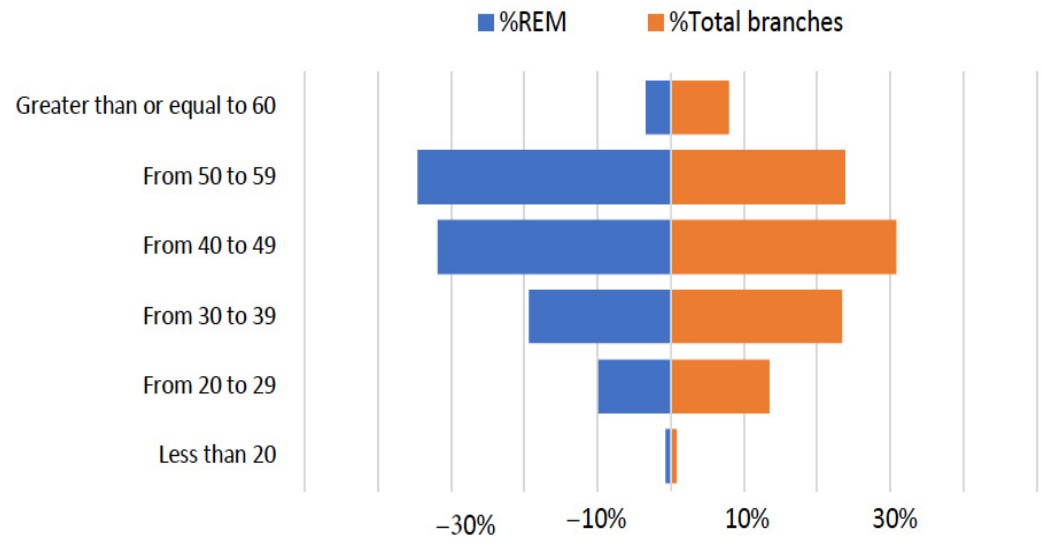

**Figure 6.** Population pyramid of the ages of workers affiliated to the REM and to the total SS branches in the Spanish Mediterranean in 2020 [9].

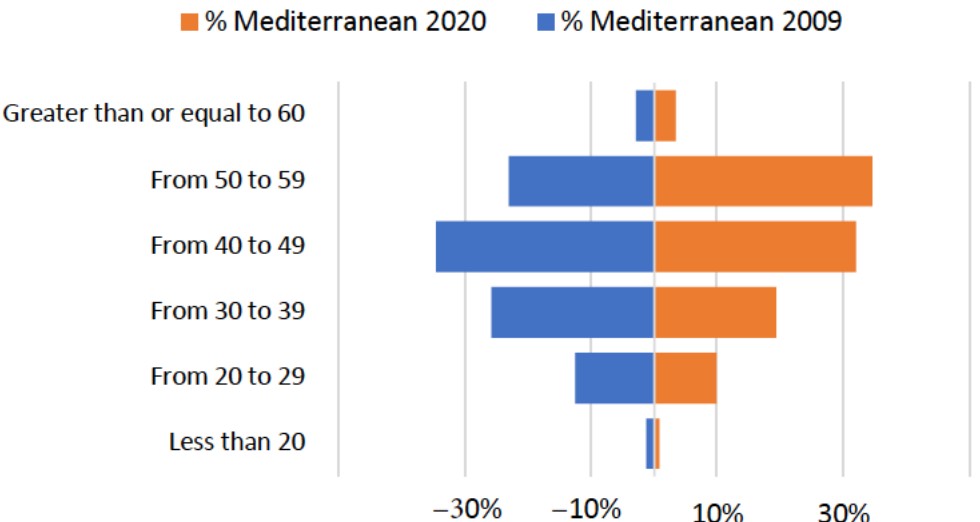

**Figure 7.** Comparison of the population pyramid of workers affiliated to the REM in the Spanish Mediterranean in 2009 and 2020 [10].

Finally, a 5-year prediction has been made for the variables of total vessels and fishers in the Spanish Mediterranean.

By performing a simple linear regression [11] for each variable, considering years as an independent variable and boats and fishers as dependent variables for each regression, two equations are obtained (Equations (1) and (2)). By applying these equations, it is possible to deduce to what extent the number of boats and crew will decrease in the future.

With the SPSS program, the following regression lines have been obtained (from annual average values 2009–2020 for fishers and 2006–2020 for boats) for the variable boats and the variable fishers.

$$\text{Boats} = 3751 - 107.85x \tag{1}$$

$$\text{Fishers} = 11{,}997 - 312.03x \tag{2}$$

As we can see in Tables 2–6, the statistics indicate that it is appropriate to apply the linear model. First of all, the value taken by the Pearson correlation or goodness coefficient is important because it reports the dependent variables, vessels and fishers are

strongly related to the time variable ($-0.96$ and $-0.97$). The negative sign both correlation coefficients hints at is, as the years go by, the number of vessels and fishers decreases. In addition, the value of R-square statistic indicates that the adjusted model explains 92% of the variability in vessels and 95% of the variability in fishers. Secondly, the *p*-value of ANOVA (analysis of variance) less than 0.05 explains that this relationship between vessels and years and between fishers and years is statistically significant, with a confidence level of 95%. It should also be noted that according to the individual significance of the variables (Tables 5 and 6), we verify that they all contribute significant information to the analysis (because *p*-value is less than 0.05).

**Table 2.** Summary of the linear regression models.

| | Model Summary | |
|---|---|---|
| | **(Fishers)** | **(Vessels)** |
| R | 0.97 | 0.96 |
| R square | 0.95 | 0.92 |

**Table 3.** ANOVA of the vessels' linear regression model.

| | ANOVA (Vessels) | | | | |
|---|---|---|---|---|---|
| | **Sum of Squares** | **df** | **Mean Square** | **F** | **Sign.** |
| Regression | 3,256,854 | 1 | 3,256,854 | 142.14 | 0.000 |
| Residual | 297,862.1 | 13 | 22,912.47 | | |
| Total | 3,554,716 | 14 | | | |

**Table 4.** ANOVA of the fishers' linear regression model.

| | ANOVA (Fishers) | | | | |
|---|---|---|---|---|---|
| | **Sum of Squares** | **df** | **Mean Square** | **F** | **Sign.** |
| Regression | 13,925,184 | 1 | 13,925,184 | 175.30 | 0.000 |
| Residual | 794,354.2 | 10 | 79,435.42 | | |
| Total | 14,719,539 | 11 | | | |

**Table 5.** The regression coefficients of the vessels' linear model estimation.

| | Coefficients (Vessels) | | | | | | |
|---|---|---|---|---|---|---|---|
| **Model** | **Unstandardized Coefficients** | | **Standardized Coefficients** | **t** | **Sign.** | **95% Confidence Interval for B** | |
| | **B** | **Std. Error** | **Beta** | | | **Lower Bound** | **Upper Bound** |
| (Constant) | 3751.00 | 82.25 | 0.00 | 45.61 | 0.000 | 3573.32 | 3928.68 |
| Years | $-107.85$ | 9.05 | $-0.96$ | $-11.92$ | 0.000 | $-127.39$ | $-88.31$ |

**Table 6.** The regression coefficients of the fishers' linear model estimation.

| | Coefficients (Fishers) | | | | | | |
|---|---|---|---|---|---|---|---|
| **Model** | **Unstandardized Coefficients** | | **Standardized Coefficients** | **t** | **Sign.** | **95% Confidence Interval for B** | |
| | **B** | **Std. Error** | **Beta** | | | **Lower Bound** | **Upper Bound** |
| (Constant) | 11,997.03 | 173.46 | 0.00 | 69.16 | 0.000 | 11,610.53 | 12,383.53 |
| Years | $-312.03$ | 23.57 | $-0.97$ | $-13.24$ | 0.000 | $-364.57$ | $-259.54$ |

The results obtained by substituting the variable years in the adjustment functions predict a decrease of 1393 fishermen in the Spanish Mediterranean from 2020 to 2025, which represents a reduction of about 17% (Figure 8). Likewise, according to the model used, 747 boats would disappear over this period, which means that by 2025, it is estimated that at this rate, there will be almost 32% fewer boats in the Spanish Mediterranean (Figure 8).

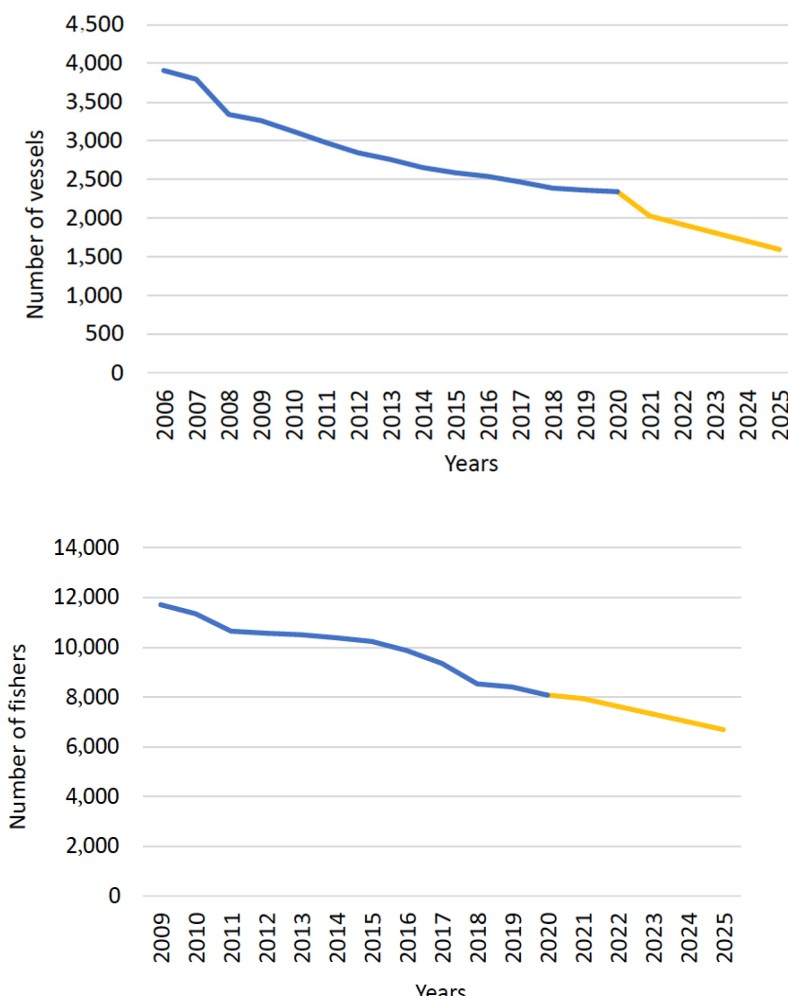

**Figure 8.** Evolution of the number of vessels and fishers [12].

## 4. Discussion

Fishing in the Spanish Mediterranean, as in many other places, especially in developed countries, is losing importance from a labor, social and economic point of view, pushed towards a certain marginality by other economic sectors such as tourism or commerce. It goes without saying that fishing is in crisis, the question now is whether this crisis will lead to the disappearance of a sector that has been present in this area for centuries. This is the subjective concern shown by the fishing sector and which this research has tried to address based on a quantitative analysis of the existing data. This question may be labeled as alarmist, but this study has tried to quantify the crisis and the results leave no room for doubt. In a period of just fifteen years, the number of vessels, power and tonnage have fallen by nearly 40%, figures similar to the decrease in landed fish. No type of fishing gear has been spared these dramatic figures, although the study seems to indicate that the main victims have been the large trawlers and the smaller ones, especially in the minor arts. The latter should be a cause for reflection, since the smaller artisanal fishing vessels, which in principle are the most selective and sustainable, are disappearing.

The fishing crisis, in its labor and social aspect, is linked to one of the biggest problems facing fishing in Europe in general and in the Spanish Mediterranean in particular, which is both a cause and a consequence of the crisis: generational replacement. The number of fishers has decreased by about 30% in fifteen years, and in just ten years the average age of fishers has increased by 3 years. Although the lack of data prevents us from drawing firm conclusions, it is a generalized impression that these numbers would be much worse if the incorporation of young immigrants had not mitigated the situation somewhat. In

fact, the role of immigrants in the Spanish (and European in general) fishing sector is a very interesting line of research which, however, can hardly be approached due to the lack of statistical data.

The analysis shows unequivocally a deep crisis reflected in a multitude of indicators. The apparently subjective concern of the sector about the current fishing situation has an obvious objective and quantifiable reasons that make us wonder what the future holds for fishing. A simple linear regression exercise based on the available data shows us an unflattering picture, with decreases in the number of fishers of around 17% (and boats of 32%) in the next 5 years. The trend therefore seems clear and difficult to reverse.

Facing the crisis of traditional fishing in the Mediterranean is therefore a question of survival for the sector. The causes of this crisis have been widely studied and debated, including overfishing, the environmental degradation of the Mediterranean Sea, the lack of profitability of the primary sectors in general and fishing in particular, the increase in labor and energy costs, the effects of fishing policies, etc.

Tackling the crisis in the fishing sector requires addressing the problems listed above. Moreover, any attempt to attract young people to the sector requires making it attractive and profitable. This seems very difficult under the current conditions. Fishing is perceived as an activity with poor working conditions, low profitability and of an unsustainable nature. The sector must face structural transformations and image overhaul: changes to avoid overexploitation of species; presenting traditional fishers as the guarantors of maritime sustainability; changes in distribution and labeling to ensure fair prices; clearly differentiating the product and presenting it as a high quality, local and sustainable product; technological changes that reduce energy and operating costs, while reducing waste and emissions; and finally, changes in fisheries policy, moving towards effective co-management that takes into account the opinions and interests of fishers.

The challenges listed above are numerous and complex, but the industry's survival is at stake. The objective of this research was to analyze the evolution of different socio-economic indicators in order to quantitatively establish the magnitude of the current crisis of the fishing sector in the Spanish Mediterranean and to establish what trend is emerging. The results are evident. The feeling of a deep crisis is not a subjective impression, it is an objective reality supported by quantitative data, which not only confirm it but also point to a trend in the same direction, at least in the next few years. This situation calls for a reflection closely linked to the issue of generational replacement by the fishing sector and public administrations. A reflection that is by no means limited to the Mediterranean. In the case of Spain, Atlantic fishing is much more powerful and has greater social, political and economic weight than Mediterranean fishing; however, the impression of a deep crisis is no less, although it may be diluted by higher overall figures. In any case, the social and labor crisis in the fishing industry is not limited to the Mediterranean or to Spain; many of the conclusions of this analysis can be extrapolated to many other (still) fishing areas of the world.

**Author Contributions:** Conceptualization, P.H.-R. and L.M.-P.; methodology, P.H.-R., J.P.G. and A.M.E.; formal analysis, A.M.E.; investigation, A.M.E., P.H.-R., J.P.G. and L.M.-P.; resources, P.H.-R.; data curation, A.M.E.; writing—original draft preparation, A.M.E.; writing—review and editing, L.M.-P. and P.H.-R. All authors have read and agreed to the published version of the manuscript.

**Funding:** This research was funded by SPANISH ECONOMY AND COMPETITIVENESS MINISTRY, grant number PID2019-105497 GB-I00, *European Fisheries Funds, opportunities for the fisheries sector through diversification, and the FLAGs management* (DivPesc).

**Institutional Review Board Statement:** Not applicable.

**Informed Consent Statement:** Informed consent was obtained from all subjects involved in the study.

**Data Availability Statement:** Publicly available datasets were analyzed in this study. This data can be found here: https://www.mapa.gob.es/es/estadistica/temas/estadisticas-pesqueras/pesca-maritima/estadistica-flota-pesquera/ (accessed on 15 September 2021).

**Acknowledgments:** Thank the support of the Spanish Marine Social Institute (ISM) in obtaining data. Thank the help of professors Roberto Cervelló and José Miguel Carot Sierra.

**Conflicts of Interest:** The authors declare no conflict of interest. The funders had no role in the design of the study; in the collection, analyses, or interpretation of data; in the writing of the manuscript; or in the decision to publish the results.

## Notes

1   On the other hand, it is expressly mentioned in the Spanish Blue Growth Strategy.

2   The term "Green Revolution" refers to the agricultural improvement program based on the application of scientific knowledge to plant breeding together with technologies that have made it possible to maximize crop yield potential.

3   Source MAPA (2006–2020) [24].

4   Source: MAPA (2006–2020) [24].

5   Source: MAPA (2006–2020) [24].

6   Source: MAPA (2004–2019) [26].

7   Source: ISM, MAPA (2009–2020) [24].

8   The retirement age for a standard worker in Spain is 67, while for a fisher it is 57.

9   Source: ISM (2020) [28]

10   Source: ISM (2009–2020) [29]

11   General expression of a straight line: y = b0 + b1x, where b0 and b1 are coefficients and the 'x' corresponds to the explanatory or independent variable, which in this case is time.

12   Source: Own elaboration. The original data for both variables are shown in blue and the values predicted from the linear regression models in yellow.

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
