# Peer review of "The Artisanal Fishing Sector in the Spanish Mediterranean: A Sector with a Long History and an Uncertain Future"

_jmse, doi:10.3390/jmse10111662_

Round 1
Reviewer 1 Report
- This works analysis and discusses the development and trends of the fishing problems in the Mediterranean. Its findings serve as a good reference for the related authorities to management and policy development and for the countries to solve the related problems.
- On page 10, the equation (1) and (2) the x need to describe in more detail and how and show how this equation was developed.
Author Response
Thank you for your constructive comments.
A footnote explaining the equation has been added.
General expression of a straight line: y=b0+b1x, where b0 and b1 are coefficients and the 'x' corresponds to the explanatory or independent variable, which in this case is time.
In addition, information on the statistics and on the reasons for choosing simple linear regression has also been added.
Reviewer 2 Report
I enjoyed reading of the manuscript as it is written in good English and the contents attractive to read. But unfortunately, I can't consider this a research or scientific manuscript. The whole paper seems to be a discussion rather than scientific writing. I couldn't find anything regarding their research in the abstract. The Introduction is lack information on the goals and objectives of their research. The materials and methods section consists of nothing regarding what methods they follow. The results section is a combination of methods and results. Unfortunately, the figures used in this manuscript were of very poor quality, and the captions were placed on top of all figures. It is unusual to say but this manuscript violated all rules and norms that are necessary for writing a research manuscript.
Author Response
Thank you for your constructive comments.
Indeed, we also had serious doubts about the scientific nature of this research. After many years of working on different economic and social aspects of fisheries, we are continually faced with enormous concern about the socio-labor situation of the sector and its immediate future. In fact, the special issue "Fisheries and Aquaculture. Current Situation and Future Perspectives", seemed to us to be the perfect place to propose this much more holistic, descriptive and general research than the ones we usually carry out.
However, it is not easy to approach the situation of the Spanish fisheries sector from an integrated point of view. The analyses, mainly of a biological and economic nature, speak of a crisis in the sector, but we know very little about the social dimension of this crisis. Indeed, the body of theoretical and practical knowledge on this aspect, which is key in fisheries sustainability programs, is relatively scarce and superficial. Advancing in this knowledge requires an initial descriptive analysis of the labor scenario. The databases, at least in Spain, are numerous, incomplete and dispersed. Therefore, a first objective of this work has been to unify and present different data that illustrate the socio-labor situation of the sector and allow us to verify and quantify a crisis that, in addition to being environmental and economic, is also social. In this way, we have tried to reach two key aspects for the future of the sector: the aging of fishermen and the lack of generational replacement.
Of course, we are aware of the descriptive nature of much of the research, so a statistical model has been proposed to try to establish a trend. However, again, the lack of data has been a problem for a more ambitious analysis. In any case, we have improved the explanation of the statistical analysis, by adding statistics, explaining the choice of simple linear regression and explaining its limitations.
Following the reviewer's suggestions, the abstract has been reworded to focus on the objective of the research; the research objectives have been expressly formulated in the introduction, in a final paragraph; in the "material and methods section" a couple of paragraphs have been added explaining the choice and characteristics of the statistical model chosen; finally, the "results section" has been reworded, moving the methodology part to the previous section.
Other questions of form, pointed out by the reviewer, have also been considered.
Reviewer 3 Report
The manuscript focus on the artisanal fishing sector in the Spanish Mediterranean. However, the content is too simple to be published in JMSE journals, and Sustainability or Fishes may be considered.
General comments:
1.There is no doubt that the authors have done a lot of work for the artisanal fishing sector of Spanish Mediterranean, but the methods and results as well as presentation and conclusion is too little. Especially, line 224-228, no basic R-squared and P-value. In addition, Boats and Fishers are influenced by many factors, e.g. energy price and labor wage. Therefore, the prediction is cautious. If this prediction is to be carried out, time series methods, such as ARMA and exponential model should be considered and used.
Specific comments:
2.Abstract section: relevant analysis results (simple introduction and significance, results, and conclusion) need to be presented, and the content is too little overall.
3.Line 245: Such studies exist and may not be in the Mediterranean region.
4.It is suggested that the author enrich the manuscript and adopt more reasonable methods to confirm the views in discussion.
Author Response
Thank you for your constructive comments.
Indeed, we also had serious doubts about the descriptive and, as the reviewer indicates, too simple nature of this research. However, after many years working on different economic and social aspects of fisheries, we felt that it was time to undertake a research that, although, a priori, simple, we believe is fundamental to understand the crisis of the sector in all its aspects. In fact, the JMSE special issue "Fisheries and Aquaculture. Current Situation and Future Perspectives", seemed to us to be the perfect place to propose this much more holistic, descriptive and general research than the ones we usually carry out.
However, it is not easy to approach the situation of the Spanish fisheries sector from an integrated perspective. The analyses, mainly of a biological and economic nature, speak of a crisis in the sector, but we know very little about the social dimension of this crisis. Indeed, the body of theoretical and practical knowledge on this aspect, which is key in fisheries sustainability programs, is relatively scarce and superficial. Advancing in this knowledge requires an initial descriptive analysis of the labor scenario. The databases, at least in Spain, are numerous, incomplete and dispersed. Therefore, a first objective of this work has been to unify and present different data that illustrate the socio-labor situation of the sector and allow us to verify and quantify a crisis that, in addition to being environmental and economic, is also social. In this way, we have tried to reach two key aspects for the future of the sector: the aging of fishermen and the lack of generational replacement. Of course, we are aware of the descriptive nature of much of the research, so a statistical model has been proposed to try to establish a trend. However, again, the lack of data has been a problem for a more ambitious statistical analysis. In any case, we believe that his suggestions as reviewer were correct and we have tried to incorporate them, both in the methodology and results sections.
The choice of simple linear regression over other possible analyses has been justified. The limitations of the model have also been emphasized; the equation and the variables have been explained in greater detail; and several tables with statistics explaining the validity of the model have been added.
Following the suggestion of the reviewers, the abstract has been modified.
Round 2
Reviewer 2 Report
I have just finished the review of the manuscript. I would like to thank the authors since they responded to every concern I raised in the previous version of my review. I think they improve the manuscript significantly and the manuscript is ready to publish now.
Reviewer 3 Report
Authors has made some modifications, and I also believe that this work is meaningful, but, this is a review manuscript, and simple content and presentation (e.g., there are very few references, too many descriptive language, but not reviewing.). I suggest that the manuscript attempts to transfer to other journals, e.g., Fishes.